# Cryopreservation of Roughscale Sole (*Clidoderma asperrimum*) Sperm: Effects of Cryoprotectant, Diluent, Dilution Ratio, and Thawing Temperature

**DOI:** 10.3390/ani12192553

**Published:** 2022-09-24

**Authors:** Irfan Zidni, Hyo-Bin Lee, Ji-Hye Yoon, Jung-Yeol Park, Hyun-Seok Jang, Youn-Su Cho, Young-Seok Seo, Han-Kyu Lim

**Affiliations:** 1Department of Biomedicine, Health & Life Convergence Science, BK21 Four, Mokpo National University, Muan 58554, Korea; 2Department of Fisheries, The Faculty of Fisheries and Marine Science, Universitas Padjadjaran, Sumedang Regency 45363, Indonesia; 3Department of Marine and Fisheries Resources, Mokpo National University, Muan 58554, Korea; 4Department of Fishery Biology, Pukyong National University, Busan 48512, Korea; 5Fisheries Resources Institute, Yeongdeok 36405, Korea

**Keywords:** cryoprotective medium, roughscale sole, sperm motility, cell survival rate, DNA damages

## Abstract

**Simple Summary:**

The roughscale sole, *Clidoderma asperrimum*, is found in the East and West Seas of Korea, waters north of Hokkaido in Japan, as well as the East China Sea, the East Pacific, and the Canadian Maritimes. In 2021, this fish was categorized as an endangered species. Thus, there is a need to maintain gametes by freezing sperm. This study investigated the impacts of the cryoprotective agent, diluent, dilution ratio, and thawing temperature on the cryopreservation of fish sperm. In this investigation, sperm dilution 1:1 with a mixture of 10% dimethyl sulfoxide + Stein’s solution and thawing at 10 °C provided the most effective DNA damage prevention. These results support the development of a roughscale sole sperm cryopreservation procedure.

**Abstract:**

The roughscale sole, *Clidoderma asperrimum* is categorized as an endangered species. Sperm freezing is essential for preserving gametes. This study examined the CPA concentration, diluent, dilution ratio, and thawing temperature to design a sperm cryopreservation protocol for roughscale sole. The variables examined included sperm motility and kinematics, cell survival, fertilization, and DNA fragmentation. Sperm motility parameters were assessed via computer-assisted sperm analysis using a CEROS II instrument. Cell survival rate and DNA damage were assessed using the Cell Counting Kit-8 and single-cell gel electrophoresis assay, respectively. Sperm preservation was tested using several CPAs, including ethylene glycol, dimethyl sulfoxide (DMSO), glycerol, propylene glycol, and methanol. The diluents tested were 300 mM sucrose, 300 mM glucose, Stein’s solution, Ringer’s solution, and Hank’s solution. The optimal conditions for sperm cryopreservation were 10% DMSO + Stein’s solution. After thawing, sperm motility was highest with a 1:1 dilution ratio (sperm to CPA + diluent), at 69.20 ± 0.32%; thawing at 10 °C was optimal for post-thaw motility (72.03 ± 0.95%). The highest fertilization rate (40.00 ± 1.22%) was obtained using DMSO. The fresh sperm had the lowest tail DNA, followed by 10% DMSO + Stein’s solution. The developed cryopreservation methods can be used in roughscale sole hatcheries.

## 1. Introduction

Juvenile development techniques for various marine fishes have been established in South Korea to support commercial production. Roughscale sole (*Clidoderma asperrimum*) is a high-quality fish in the flounder family that has good meat quality for sashimi, with a light and elastic character [1]. This fish lives in many places, including the East and West Seas of Korea; north of Hokkaido, Japan; the East China Sea; Canada; and the East Pacific [2]. This species lives in muddy sand bottoms at depths of 150–1000 m in the coastal waters of Korea and Japan. It can grow up to 62 cm in length and weigh up to 4.4 kg. In nature, catch of this species is low; therefore, increased production through aquaculture is needed. A stable juvenile supply is essential to the aquaculture of this species, but many difficulties must be overcome; no juvenile production technology has yet been established for roughscale sole. Limited sperm production in mature broodstock of this species during the artificial production of juveniles is a problem.

Sperm cryopreservation is a common practice in the aquaculture sector, with numerous benefits. In industrial fish hatcheries, sperm cryopreservation is a practical method for simple and efficient broodstock control. Sperm cryopreservation has numerous benefits, including gamete accessibility, conservation of genetic information, simplified sperm transport, reduced disease transmission, and lower cost of raising male broodstock. Furthermore, because of the insufficient number of gonadally mature males in the hatchery system, sperm cryopreservation offers a feasible option for artificial fertilization. Cryopreservation studies have examined other flatfish species, but the cryopreservation of sperm from roughscale sole has not yet been investigated.

For sperm cryopreservation, there is a need to understand gamete physiology and the biochemical processes that occur when sperm are collected, cooled, processed, and thawed [3]. Specifically, the composition of the cryomedium, composed of a cryoprotective agent (CPA) and a diluent, is a key consideration [4]. The CPA protects cells from injury caused by freezing and thawing, whereas the diluent helps to maintain the dormant state of sperm cells and support their metabolic processes [5]. Therefore, the optimal diluent for the cryopreservation of sperm from a particular species is selected based on the characteristics of the sperm. Another factor that affects the quality of spermatozoa is the cooling rate, which greatly impacts the motility of frozen and thawed sperm. Other factors that influence the cryopreservation of fish sperm include the dilution ratio, which is associated with physical shock during dilution, and the thaw temperature, which is associated with rapid and full restoration of membrane integrity and permeability.

Because of the species-specific impacts of cryomedium composition, research is needed to determine the type and concentration of CPA, as well as the optimal diluent for roughscale sole sperm cryopreservation. The main objective of our study is to establish an ideal cryopreservation protocol for this species, including selection of the CPA, diluents, dilution ratio, and thawing rates, all of which are key factors for successful cryopreservation.

## 2. Materials and Methods

### 2.1. Fish Handling and Sperm Collection

The experiment was conducted at the Fisheries Resources Institute, Yeongdeok, Gyeongsangbuk-do, South Korea. Male roughscale sole (*n* = 9) with mean length of 35.28 ± 0.61 cm and weight of 526 ± 29.82 g were kept isolated in seawater flow, at temperature of 10–11 °C, salinity of 32 ± 1.50 psu, dissolved oxygen of 8.62–9.21 mg/L, and pH of 7.82–7.90. The fish were fed a commercial diet (Merk, Super Plus 7S, Korea) once daily during the study. After acclimatization, tricaine methanesulfonate (MS-222) was applied to anesthetize the fish and spermiation was stimulated through intramuscular injection of Ovaprim (0.3 mL kg^−1^; Syndel Laboratories, Nanaimo, BC, Canada) [6,7]. After 3 days of hormonal treatment, sperm samples were obtained by abdominal stripping and direct collection using 1.0-mL plastic syringes. To prevent contamination with excrement, water, blood, mucus, or urine, the region surrounding the male genitalia was sanitized [8]. The collected sperm were placed into a box on ice and directly transported to the laboratory for analysis, observation, and cryopreservation. In this study, sperm samples with total motility >80% were pooled for use. The sperm volume of the samples ranged from 0.2–0.8 mL/fish, whereas the sperm count was approximately 98,000,000–226,000,000 mL^−1^.

### 2.2. Cryoprotective Medium

For the first experiment, several CPAs, including ethylene glycol (EG), dimethyl sulfoxide (DMSO), methanol, propylene glycol (PG), and glycerol, were used to examine the influence of CPA selection on the cryopreservation of sperm from roughscale sole. All CPAs were applied at a final concentration of 10%. In this study, 300 mM sucrose was used as the diluent. The resulting combined CPA and diluent solution was added to sperm at a 1:1 ratio in a 1.5-mL tube. No equilibration period was used. The solution was transferred to 0.25-mL cryopreservation straws (Classic IMV Technologies, L’Aigle, France), which were sealed with sealing powder (Reproduction Provisions, Walworth, WI, USA). In this study, two-step freezing was performed. In the first step, each altered sperm sample was placed on a floating device composed of a tray within an expanded polystyrene box at a height of 3 cm above liquid nitrogen for 2 min, and a lid was used to maintain constant temperature during the initial cooling phase; in the second step, the sperm sample was submerged in liquid nitrogen (196 °C) [9]. The control treatment was fresh sperm obtained from the same male. Each group was assessed using three replicates for this analysis, and the straws were thawed in distilled water at 10°C prior to use [4]. An additional experiment was conducted to investigate the influence of diluent type using 300 mM sucrose, 300 mM glucose, Stein’s solution (NaCl 0.75 g, KCL 0.038 g, Hen egg yolk 100 mL, C_6_H_12_O_6_ 0.10 g, and NaHCO_3_ 0.20 g L^−1^ distilled water), Ringer’s solution (NaCl 13.50 g, KCL 0.60 g, CaCl_2_ 0.35 mL, NaHCO_3_ 0.03, and MgCl_2_ g L^−1^ distilled water), and Hank’s solution (NaCl 8.0 g, KCL 0.40 g, CaCl_2_ 0.14 mL, MgSO4.7H_2_O 0.06 g, KH_2_PO_4_ 0.03, C_6_H_12_O_6_ 0.10 g and NaHCO_3_ 0.17 g L^−1^ distilled water). The CPA for this investigation constituted 10% DMSO. Each mixture was frozen and thawed in the same manner as in the first experiment. The most effective CPA was identified in the first experiment; therefore, the third experiment was conducted to determine the optimal concentration of DMSO. DMSO was tested at various concentrations (5%, 10%, 15%, and 20%) with Stein’s solution as the diluent. The freezing and thawing procedure was performed as in the previous experiment.

### 2.3. Dilution Ratio and Thawing Temperature

Previous investigations demonstrated that a mixture of 10% DMSO and Stein’s solution is optimal for the cryopreservation of sperm from roughscale sole. To assess the effect of the dilution ratio, sperm samples were diluted 1:1, 1:3, 1:5, 1:10, and 1:100 with Stein’s solution containing 10% DMSO. To determine the ideal thawing temperature, straws were retrieved from liquid nitrogen storage and immersed for 15 s in water at 5 °C, 10 °C, 15 °C, and 20 °C.

### 2.4. Assessment of Sperm Motility

Sperm samples were evaluated via computer-assisted sperm analysis with a CEROS II instrument (Hamilton Thorne, Inc., Beverly, MA, USA). Briefly, 10 µL of each sperm sample were diluted with artificial seawater (containing 27 g NaCl, 0.5 g KCl, 1.2 g CaCl_2_, 4.6 g MgCl_2_, and 0.5 g NaHCO_3_ per liter of distilled water stored in 4 °C) at a ratio of 1:99. Activation was observed in a leja slide -10 µm (IMV Technologies, France) using a microscope at 10× magnification (Zeiss Axiolab 5) connected to a computer with a CM-040GE camera (JAI, Japan) with 0.4-megapixel resolution at 60 frames per second; the sperm samples were assessed in 10 °C room temperature using the computer-assisted sperm analysis instrument CEROS II with counting a minimum of 200 spermatozoa. Several kinematic parameters were investigated including the proportion of motile sperm in percentage form (motility); curvilinear velocity (VCL), the average speed of a sperm cell moving along a curved path; straight-line velocity (VSL), the average speed of a sperm cell moving along a straight line; path velocity (VAP), the distance traveled by a sperm over time along a five-point mathematically smoothed path; percent straightness (STR), straight-line velocity (VSL) as a proportion of VAP in percentage form; and percent linearity (LIN), the ratio of VSL to VCL [10]. Additionally, the quality of cryopreserved sperm was observed 15 s after thawing in a 10 °C water bath. Each treatment was tested three times with three sets of experiments.

### 2.5. Evaluation of Survival Rate

Using Cell Counting Kit-8 (CCK-8) assays (Bimake, Houston, TX, USA), the survival rates of fresh and thawed sperm cells were evaluated in accordance with a previously described method [11]. CCK-8 reagent was added directly to cells in culture medium at a volume ratio of 1:10. In the first step, cell suspensions (100 µL/well) were placed into 96-well plates. Then, 10 µL of the kit reagent were poured into each well of the plate; the cells and reagent were incubated for 1–4 h until an orange color was observed. The absorbance was measured using a microplate reader at a wavelength of 450 nm (Spectra Max 190, Molecular Devices, San Jose, CA, USA). The amount of formazan (an orange dye) produced by dehydrogenase activity indicates the number of living cells. In this test, orange sperm are alive, whereas white sperm are dead [7].

### 2.6. Single-Cell Gel Electrophoresis

Sperm DNA was damaged during the cryopreservation and thawing operations, even when the ideal CPA and diluent were used, as determined in a previous experiment. DNA damage was assessed using the single-cell gel electrophoresis assay (Comet Assay, Trevigen, Gaithersburg, MD, USA). Fluorescence labeling with SYBR Gold (diluted 10,000-fold in DMSO) was performed 30 min after the assay; a fluorescence microscope was used to observe the results (DM 2500 microscope, Leica, Wetzlar, Germany). The Comet Assay IV lite System was used to measure head length: the diameter of the nucleus, head intensity (DNAh): the mean pixel intensities in the head [12], tail length: the distance between the center of the nucleus and the most distant point of DNA migration [13], tail intensity: the mean pixel intensity in the tail of the comet [14], and tail migration (Instem, Staffordshire, UK). The proportion of DNA from the tail was determined using the following formula: [100 × (DNAc − DNAh)/DNAc] [15,16], where DNAc and DNAh represent the summed intensities of the pixels in the entire comet and the head region, respectively [17]. The levels of damage were classified and the proportion of cells within each category was determined [8].

### 2.7. Fertilization Rate Evaluation

The analysis of fertilization was performed at the Fisheries Resources Institute in Gyeongsangbuk-do. The eggs used in the experiments were from *C. asperrimum* (*n* = 3, length: 41.53 ± 1.60 cm, weight: 1176.67 ± 195.21 g) and the fertility rate in each experiment was determined by offspring production. The sperm to eggs ratio was 0.25 mL: 5 mL (1:20) and the duration of fertilization was 2 min; three washes were then performed. For each experiment, 500 fertilized eggs were randomly selected and allowed to settle at 18 °C; this process was repeated three times. The fertilized eggs were counted after 4 cell cycles, using a microscope at 40× magnification (CH30, Olympus, Tokyo, Japan) [11].

### 2.8. Statistical Analysis

All data are presented as means and standard errors. IBM SPSS Statistics 25.0 software (Chicago, IL, USA) was used to conduct statistical comparisons by one-way analysis of variance. The means were examined with Duncan’s multiple range test, and differences were considered statistically significant at *p* < 0.05.

## 3. Results

### 3.1. Effects of Cryoprotective Agent on post-Thaw Motility, Kinematic Parameters, and Cell Survival Rate

The optimal CPA for the cryopreservation of sperm from roughscale sole was initially investigated using DMSO, PG, EG, methanol, and glycerol. The collected data showed that the CPA significantly (*p* < 0.05) affected post-thaw motility, kinematic parameters, and cell survival rate (Figure 1). When 300 mM sucrose was used as the diluent, the most effective post-thaw motility was much greater with 10% DMSO than with other CPAs, at 56.48 ± 1.20%. Additionally, 10% PG and 10% EG performed well as CPAs, with post-thaw motilities of 44.20 ± 1.99% and 38.62 ± 3.84%, respectively (*p* < 0.05). In contrast, methanol had no effect on sperm motility when 300 mM sucrose was utilized as the diluent. For the kinematic parameters VAP, VCL, and VSL, the use of DMSO as the CPA resulted in high-quality post-thaw sperm, with values of 74.06 ± 1.83 µm/s, 95.98 ± 2.86 µm/s, and 50.51 ± 0.89 µm/s, respectively. Compared with other CPAs, the highest cell survival rate was observed for sperm cryopreserved with 10% DMSO (68.89 ± 3.18%; *p* < 0.05), followed by 10% PG (56.84 ± 2.65%; *p* < 0.05) and EG (50.67 ± 2.08%; *p* < 0.05). In contrast, sperm cryopreserved using methanol showed the lowest cell survival rate and the lowest post-thaw motility.

### 3.2. Effects of Diluent on Post-Thaw Motility, Kinematic Parameters, and Cell Survival Rate

To investigate the effects of the diluent used for cryopreservation of sperm from roughscale sole, we tested 300 mM sucrose, 300 mM glucose, Stein’s solution, Hank’s solution, and Ringer’s solution with 10% DMSO. The combination of 10% DMSO and Stein’s solution produced significantly higher post-thaw motility (63.13 ± 2.04%) than other diluents (*p* < 0.05), but this value was lower than motility in the control group (fresh sperm) (Figure 2). Additionally, 300 mM sucrose performed well as a diluent, with post-thaw motility of 38.08 ± 0.36%. For the kinematic parameters VAP, VCL, and VSL, the use of Stein’s solution as the diluent resulted in high-quality post-thaw sperm, with values of 100.72 ± 0.87 µm/s, 126.04 ± 4.62 µm/s, and 95.49 ± 0.70 µm/s, respectively (*p* < 0.05). As a diluent, Hank’s solution produced the lowest post-thaw motility, followed by 300 mM glucose and Ringer’s solution. Among diluents, Stein’s solution showed the highest percentage of surviving cells (84.78 ± 2.42%; *p* < 0.05), followed by 300 mM sucrose (73.74 ± 3.23%; *p* < 0.05).

### 3.3. Effects of DMSO Concentration on Post-Thaw Motility, Kinematic Parameters, and Cell Survival Rate

The impact of DMSO concentration was studied using Stein’s solution as the diluent. Higher post-thaw motility was observed with 10% DMSO, at 82.35 ± 1.18%; however, this value was not significantly different from the value obtained with 15% DMSO (80.48 ± 0.65%) (Figure 3). For the kinematic parameters VAP, VCL, and VSL, 10% DMSO resulted in significant differences in post-thaw sperm, with values of 85.41 ± 3.36 µm/s, 114.59 ± 2.78 µm/s, and 60.36 ± 2.75 µm/s, respectively (*p* < 0.05). In this experiment, 5% and 20% DMSO resulted in low motility and were therefore considered non-optimal DMSO concentrations. The optimal DMSO concentration depends on the type of diluent and is highly species-specific. The effect of DMSO concentration was investigated using Stein’s solution as the diluent. Higher cell survival rates were observed with 10% DMSO (81.31 ± 3.51%) compared with other concentrations, followed by 15% DMSO (69.39 ± 4.77%; *p* < 0.05).

### 3.4. Effects of Sperm Dilution Ratio on Post-Thaw Motility, Kinematic Parameters, and Cell Survival Rate

The effects of the ratio of sperm to diluent (i.e., sperm dilution ratio) on the quality of cryopreserved sperm are shown in Figure 4. The highest post-thaw motility was observed with a ratio of 1:1, at 69.20 ± 0.32%, followed by a 1:3 ratio, at 55.63 ± 2.28%. In contrast, no sperm motility was observed at the non-optimal dilution ratios of 1:5 to 1:100. For the kinematic parameters VAP, VCL, and VSL, the use of a 1:1 ratio resulted in significant differences in post-thaw sperm, with values of 74.06 ± 1.83 µm/s, 95.98 ± 2.85 µm/s, and 50.50 ± 0.89 µm/s, respectively (*p* < 0.05). Higher cell survival rates were observed at a dilution ratio of 1:1 (81.15 ± 2.40%) compared with other dilution ratios, followed by 1:3 (54.58 ± 3.52%).

### 3.5. Effects of Thawing Temperature on Post-Thaw Motility, Kinematic Parameters, and Cell Survival Rate

Various thawing temperatures (5 °C, 10 °C, 15 °C, and 20 °C) were used after sperm had been frozen in a liquid nitrogen tank (−196 °C). The impacts of thawing temperature on the motility of cryopreserved sperm from roughscale sole are shown in Figure 5. The highest sperm motility was observed after thawing at 10 °C for 15 s (72.03 ± 0.95%), followed by thawing at 15 °C (67.33 ± 1.80%). The highest VAP, VCL, and VSL values were observed after thawing at 10 °C for 15 s, with values of 74.06 ± 1.83 µm/s, 95.98 ± 2.85 µm/s, 50.50 ± 0.89 µm/s, respectively. The highest cell survival rate was observed for cryopreserved sperm thawed at 10 °C (59.32 ± 2.07%; *p* < 0.05) compared with sperm thawed at other temperatures.

### 3.6. Effect of Treatment Method on DNA Damage

To determine the extent of DNA damage caused by cryopreservation, we assessed the effects of various cryoprotective agents on sperm via single-cell gel electrophoresis with 300 mM sucrose as the diluent. The lowest tail length, tail intensity, percentage of tail DNA, and tail migration were obtained in the control group (fresh sperm), followed by the DMSO treatment, which had lower values than other CPAs (*p* < 0.05, Table 1). Comet assay images for this treatment group are showed in Figure 6.

In contrast, glycerol and methanol produced greater tail length, tail intensity, and tail migration. The impacts of diluent on cryopreservation-induced damage were studied using 10% DMSO as the CPA. Treatment with Stein’s solution produced low tail intensity, tail length, percentage of tail DNA, and tail migration values compared with other treatments, but these values were higher than values in the control group (*p* < 0.05; Table 2). Figure 7 displayed images of the comet assay for this treatment group.

Furthermore, treatment with Hank’s solution and 300 mM glucose led to higher values for tail intensity, percentage of tail DNA, tail length, and tail migration. To identify any cryopreservation-induced DNA damage, we tested various concentrations of DMSO using Stein’s solution as the diluent. The lowest tail length, tail intensity, percentage of tail DNA, and tail migration values were observed in the control group (fresh sperm), followed by the 10% DMSO treatment, which had values lower than other DMSO concentrations (*p* < 0.05, Table 3). In contrast, 20% DMSO resulted in higher values for the percentage of tail DNA and tail migration. Figure 8 showed comet assay images for this treatment group.

### 3.7. Effect of Treatment Method on Fertilization Rate

Eggs from mature females were pooled for investigation of the fertilization rate. Each experiment utilized a dry technique for artificial insemination. Fertilization analysis indicated that fresh sperm had the highest fertilization rate (42.90 ± 1.56%) among the tested treatments (Figure 9). Investigation of various CPAs showed that the highest fertilization rate was achieved with DMSO (40.00 ± 1.22%); this rate was not significantly different from the rate exhibited by fresh sperm. Among diluents, the highest fertilization rate was achieved with Stein’s solution (38.8 ± 0.92%). The fertilization rates obtained using various DMSO concentrations indicated that 10% DMSO supported significantly higher fertilization (33.3 ± 1.10%) compared with other concentrations of DMSO. The dilution ratio of 1:1 led to the highest fertilization rate (35.20 ± 0.75%); fertilization was also optimal with thawing rates of 10 °C and 15 °C. Generally, the results of fertilization analysis were consistent with the results of post-thaw motility and cell survival analyses in all treatment methods.

## 4. Discussion

The development of an effective sperm cryopreservation technique is essential for preserving biodiversity, reducing inbreeding, and minimizing domestic selection. Multiple factors influence the quality of cryopreserved sperm including cryomedium composition, freezing speed, equilibration period, dilution ratio, and thawing temperature [18]. For the cryopreservation of fish sperm, the most important variables are the CPA and diluent [19]. In this study, 10% DMSO was the most effective CPA for the cryopreservation of sperm from roughscale sole, *C. asperrimum*; it produced the highest rates of sperm movement and cell survival after thawing. DMSO was previously identified as an effective CPA for fish sperm cryopreservation [20]. Some flatfish species have exhibited great frozen–thawed sperm quality when DMSO was used. In starry flounder, *Platichthys stellatus*, 10% DMSO was an effective CPA for sperm cryopreservation [21]; in summer flounder, *Paralichthys dentatus*, 15% DMSO was effective for sperm cryopreservation [22]. For sperm cryopreservation in Brazilian flounder, DMSO resulted in the highest quality of frozen sperm [23]; 10% DMSO was optimal for the cryopreservation of sperm from turbot, *Scophthalmus maximus* [24]; 7.5% to 10% DMSO was effective for cryopreservation of sperm from stone flounder, *Kareius bicoloratus* [16]; for cryopreservation of sperm from the spotted halibut, *Verasfer variegatus*, 15% DMSO produced high-quality frozen sperm [4]; and also 15% DMSO was the optimal concentration for cryopreservation of sperm from marbled flounder, *Pseudopleuronectes yokohamae* [25]. Thus, the optimal concentrations of DMSO as a cryoprotectant for freezing fish sperm are between 5% and 25% [26]. DMSO can pass through cellular membranes and slow the flow of water from cells toward ice crystals outside of cells, lower the temperatures of gaseous bubbles and ice crystals, and reduce ice crystal size [27]. DMSO is a small polar molecule; compared with other CPAs, it can readily pass through cellular membranes via passive diffusion [28]. CPAs also provide protection by regulating the pace at which a cell dehydrates during freezing through reduction of the energy barrier needed to move water across the membrane [29]. To minimize physical and chemical damage during the freezing and thawing of fish sperm, cryoprotective agents must exhibit low molecular weight, neutrality, and hydrophilicity. Additionally, such agents must function as a solvent for electrolytes or organic acids; they must also exhibit high cellular permeability and low cellular toxicity. Because each CPA has a low eutectic point, it quickly penetrates cells, inhibits dehydration, and prevents changes to the colloidal state of the protoplasm by reducing the formation of ice crystals; thus, it minimizes sperm damage after freezing and thawing [9,30]. Based on the results of the first experiment in the present study, the use of DMSO led to the best cryopreservation of sperm from roughscale sole. In contrast, the use of methanol and glycerol led to low post-thaw roughscale sole sperm quality. Similar results were obtained for the cryopreservation of flounder sperm, including starry flounder, summer flounder, turbot, stone flounder, spotted halibut, and marbled flounder—the use of methanol or glycerol was ineffective. Generally, methanol was effective in cryopreservation of sperm from freshwater fish, including zebrafish, *Danio rerio* [30]; cyprinid fish, *Neolissochilus soroides* (Valenciennes, 1842) [31]; black-stripe black crappie, *Pomoxis nigromaculatus* [32]; goldfish, *Carassius auratus* [33]; Japanese bitterling [34]; and masu salmon, *Oncorhynchus masou* [35]. In contrast, glucose-methanol has been employed for the cryopreservation of sperm from Mozambique tilapia, *Oreochromis mossambicus* [36]. Moreover, glycerol was effective for the cryopreservation of sperm from several species, including giant grouper [9], pointhead flounder (*Cleisthenes pinetorum herzensteini*) [37], and longtooth grouper (*Epinephelus bruneus*) [38].

In addition to the CPA, the diluent acts as a medium for cryopreservation and contains compounds that prevent cellular injuries associated with intracellular ice crystal formation [27]. In the present study, Stein’s solution produced higher motility in frozen–thawed sperm, compared with 300 mM sucrose, 300 mM glucose, Hank’s solution, and Ringer’s solution. In a previous study, Stein’s solution was effective as the diluent in cryopreservation of sperm from starry flounder [21]. Stein’s solution has also been effectively used for the cryopreservation of sperm from some freshwater teleosts [39]. Diluents are used to dilute fish milt prior to freezing; they are designed to be compatible with the seminal plasma of fish [23]. Therefore, diluents have essential roles in cryopreservation. Diluent selection for cryopreservation is important for multiple reasons because the diluent may provide conditions that actively restrict sperm survival, prevent the activation of sperm due to changes in osmotic pressure during dilution, or increase the number of viable sperm. The composition of Stein’s solution includes NaCl, KCl, hen egg yolk, glucose, and NaHCO_3_. According to Suquet et al. [40], most diluents used for marine fish sperm are solutions of saline (concentration 1–10%) or sugar (5–10%). Sugar provides sperm with a glycolyzable substrate, avoids agglutination, maintains osmotic tension and electrolyte balance, and exhibits cryoprotective effects during freezing [27]. Sucrose was the second most effective diluent, after Stein’s solution. The application of sucrose as a diluent has been reported in kelp grouper, *Epinephelus moara* [41]; giant grouper [11]; spotted halibut [4]; Brazilian flounder [23]; winter flounder, *Pseudopleuronectes americanus* [42]; and summer flounder [43]. In contrast, the use of Hank’s solution, 300 mM sucrose, and Ringer’s solution resulted in low post-thaw sperm quality. Generally, previous research has shown that Hank’s solution is effective for the cryopreservation of sperm from freshwater fish, including green swordtail [44], stinging catfish, and Philippine catfish [45].

Cryopreservation efficiency may be influenced by the dilution ratio. In the present study, higher post-thaw motility was observed when a 1:1 ratio was used. The VCL, VAP and VSL parameters significantly increased, and the cell survival rate was also high with a 1:1 dilution ratio. Increasing the dilution ratio from 1:3 to 1:100 lowered the quality of cryopreserved sperm. In this study, the dilution ratio had a significant impact on cryopreserved sperm motility, velocity, and cell survival rate, indicating that a higher dilution ratio is associated with lower post-thaw sperm quality. In previous research, the cryopreservation of sperm from flounder species showed a range of optimal dilution ratios, including 1:2 in summer flounder [43] and 1:3 in spotted halibut, Brazilian flounder, and starry flounder [4,21,23]. The cryopreservation of stone flounder sperm with dilution ratios between 1:1 and 1:10 resulted in a higher proportion of motile sperm [16]. The proportion of motile frozen–thawed turbot spermatozoa was unaffected by increasing the dilution rate from 1:1 to 1:9 [46]. The optimal dilution ratio for cryopreservation of fish sperm is frequently presented as a range of species-specific values ranging from 1:1 to 1:20 [47], with higher dilution ratios generally resulting in lower percentages of post-thaw motile sperm. This trend was validated by the work of Correa and Zavos [48], who determined that rapid changes in osmotic pressure caused osmotic shock when diluents were applied. Sperm viability during the dilution process may be negatively affected by a high sperm dilution rate. When the concentration of the CPA + diluent mixture is greater than the concentration of seminal plasma, fish sperm can be exposed to rapid environmental changes.

Physiological processes during thawing are similar to the processes during freezing, but the order is reversed. Theoretically, the rates of thawing and cooling should be identical. To avoid sperm destruction, there is a need to prevent additional stress caused by irregular temperature variations. In the present study, the quality of cryopreserved sperm was highest after thawing at a temperature of 10 °C. In a previous investigation, the quality of cryopreserved sperm was excellent when a temperature of 10 °C was used for Atlantic halibut [49,50] and when a temperature of 10.5 °C was used for spotted halibut [4]. Ideal thawing temperatures vary among marine species. The optimal temperatures for other flounders include 20 °C for starry flounder and stone flounder [16,21], 37 °C for Brazilian flounder and Atlantic halibut [15,23], 35 °C for summer flounder [22], 30 °C for winter flounder [42], and 40 °C for turbot [51]. The quality of fish sperm diminishes with repeated cooling and warming; it is also influenced by the adaptations of each fish species. According to Batsy and Kumar et al. [27], the time and temperature of thawing should be based on the package dimensions, form, and composition, as well as the thawing medium. The duration of thawing is generally 30 s for mini-straws and 5 min for cryovials.

The evaluation of DNA damage in cryopreserved sperm is essential for observing genetic problems and also for level of fertilizing ability prediction [15]. In the control group (fresh sperm), the tail DNA level was lowest, followed by treatment with 10% DMSO + Stein’s solution; these levels significantly differed from the levels in other treatments. In a previous study, we evaluated DNA damage in cryopreserved sperm from spotted halibut and stone flounder. Similar to the present results, we found that the use of DMSO for sperm cryopreservation resulted in lower DNA damage values than when other CPAs were used; however, these values were higher than the values in than control group (fresh sperm). Concerning the cryopreservation of other fish species, DNA damage has also been observed in sea bass, *Dicentrarchus labrax* [52]; rainbow trout, *Oncorhynchus mykiss*; gilthead sea bream, *Sparus aurata* [15,53]; *Sparus macrocephalus* [54]; and viviparous black rockfish, *Sebastes schlegelii* [55]. DNA damage in frozen fish sperm is affected by the CPA, diluent, fish species, and chromosomal structure [26]. In the present study, we examined the effects of various CPAs, DMSO concentrations, and diluents on DNA damage. The results of the comet assay varied among CPAs, DMSO concentrations, and diluents. In a study of the cryopreservation of sperm from African turquoise killifish, *Nothobranchius furzeri*, 10% DMSO prevented cryodamage and preserved sperm viability [56]. In previous studies of DNA damage in frozen sperm, many researchers have found that it is associated with lower fertilization rates.

In this study, we examined the effects of treatment methods on the fertilization rate. The highest fertilization rate was observed with 10% DMSO and Stein’s solution as the cryomedium. These results indicate that the combination of 10% DMSO and Stein’s solution provides a better cryomedium than the other combinations tested for the cryopreservation of sperm from roughscale sole. In other species, the application of DMSO as the CPA has also resulted in high fertilization rates for black-stripe black crappie, *Pomoxis nigromaculatus* [32]; summer flounder [22]; and Brazilian flounder [23]. Effective penetration of the cryoprotectant through the cell membrane supports high fertilization rates. The effectiveness of the CPA selected for sperm cryopreservation is also influenced by factors such as fish genetic quality, sample collection procedures, and storage protocols [55]. Sperm that were cryopreserved with a 1:1 dilution ratio and thawed at 10°C showed excellent sperm quality and the highest fertilization rate.

## 5. Conclusions

Suitable conditions for the cryopreservation of sperm from roughscale sole were investigated; among the tested combinations, 10% DMSO and Stein’s solution produced frozen–thawed sperm with the highest motility, cell survival rate, and fertilization rate, as well as the minimum DNA damage. After thawing, high levels of motility and kinematic parameters were detected with a dilution ratio of 1:1. Thawing at 10 °C for 15 s was the most effective technique for preserving the quality of frozen–thawed sperm.

## Figures and Tables

**Figure 1 animals-12-02553-f001:**
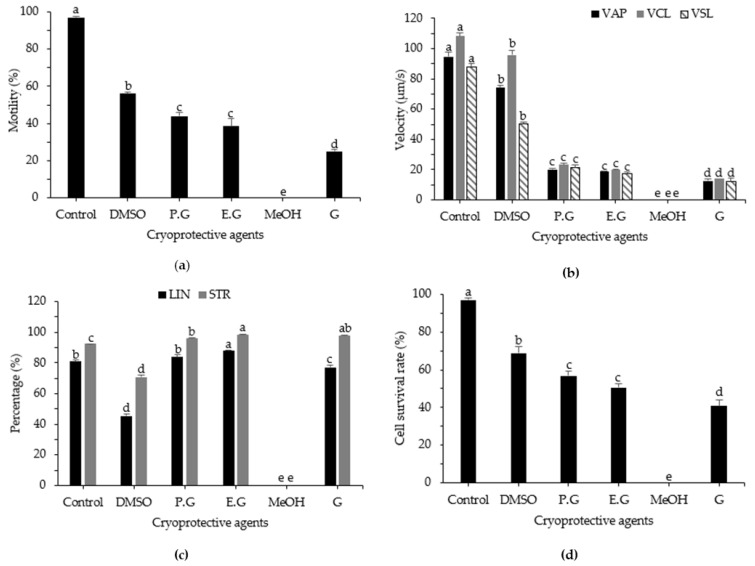
Computer-assisted sperm analysis of frozen and fresh sperm from roughscale sole, *Clidoderma asperrimum*, using various cryoprotective agents with 300 mM sucrose. (**a**) Motility (%), (**b**) average path velocity (VAP; μm s^−1^), curvilinear velocity (VCL; μm s^−1^), straight-line velocity (VSL; μm s^−1^), (**c**) linearity (LIN; %), straightness (STR; %), and (**d**) cell survival rate (%). Different lowercase letters indicate significant differences among cryoprotective agents (*p* < 0.05). Control, fresh sperm; DMSO, dimethyl sulfoxide; EG, ethylene glycol; PG, propylene glycol; MeOH, methanol; G, glycerol.

**Figure 2 animals-12-02553-f002:**
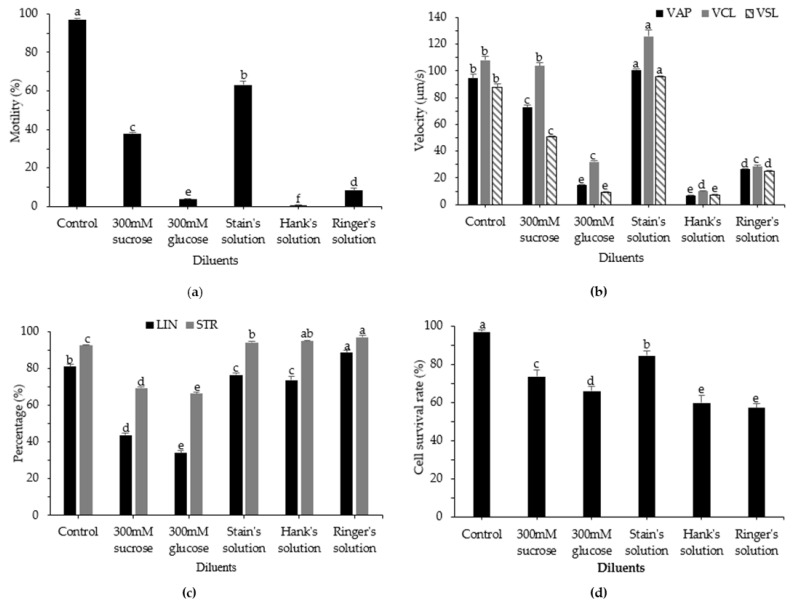
Computer-assisted sperm analysis of frozen and fresh sperm from roughscale sole, *Clidoderma asperrimum*, using various diluents with 10% DMSO as the cryoprotectant. (**a**) Motility (%), (**b**) average path velocity (VAP; μm s^−1^), curvilinear velocity (VCL; μm s^−1^), straight-line velocity (VSL; μm s^−1^), (**c**) linearity (LIN; %), straightness (STR; %), and (**d**) cell survival rate (%). Different lowercase letters indicate significant differences among diluents (*p* < 0.05). Control, fresh sperm.

**Figure 3 animals-12-02553-f003:**
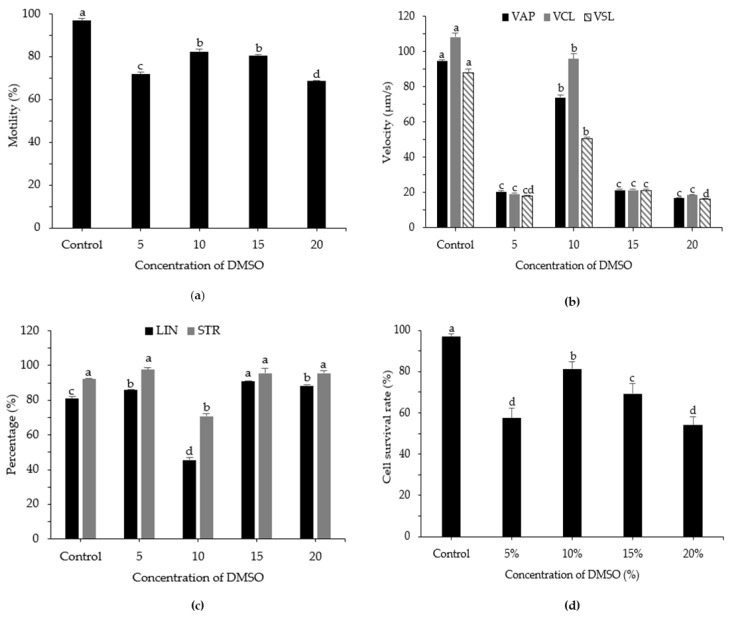
Computer-assisted sperm analysis of frozen and fresh sperm from roughscale sole, *Clidoderma asperrimum*, using various concentrations of DMSO with Stein’s solution as the diluent. (**a**) Motility (%), (**b**) average path velocity (VAP; μm s^−1^), curvilinear velocity (VCL; μm s^−1^), straight-line velocity (VSL; μm s^−1^), (**c**) linearity (LIN; %), straightness (STR; %), and (**d**) cell survival rate (%). Different lowercase letters indicate significant differences among concentrations of DMSO (*p* < 0.05). Control, fresh sperm.

**Figure 4 animals-12-02553-f004:**
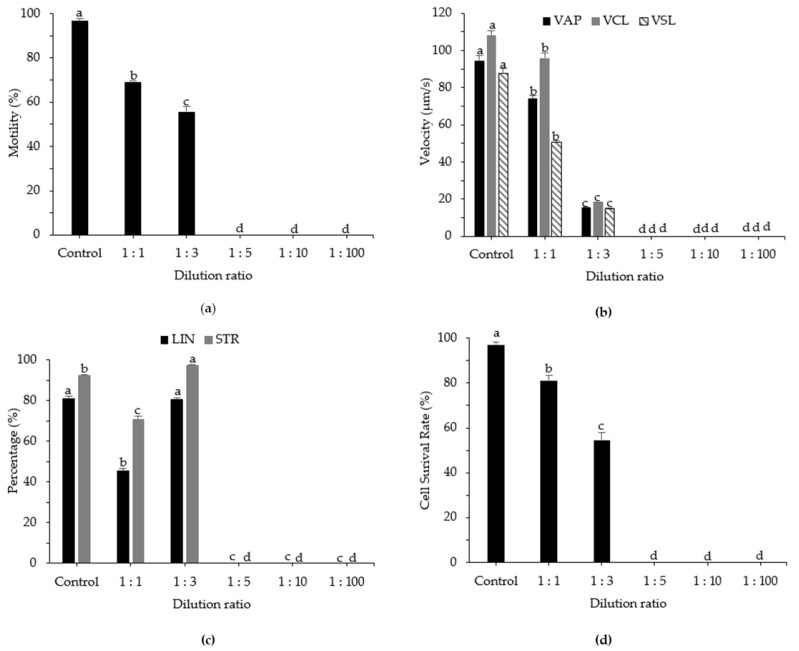
Computer-assisted sperm analysis of frozen and fresh sperm from roughscale sole, *Clidoderma asperrimum*, at various dilution ratios. (**a**) Motility (%), (**b**) average path velocity (VAP; μm s^−1^), curvilinear velocity (VCL; μm s^−1^), straight-line velocity (VSL; μm s^−1^), (**c**) linearity (LIN; %), straightness (STR; %), and (**d**) cell survival rate (%). Different lowercase letters indicate significant differences among dilution ratios (*p* < 0.05). Control, fresh sperm.

**Figure 5 animals-12-02553-f005:**
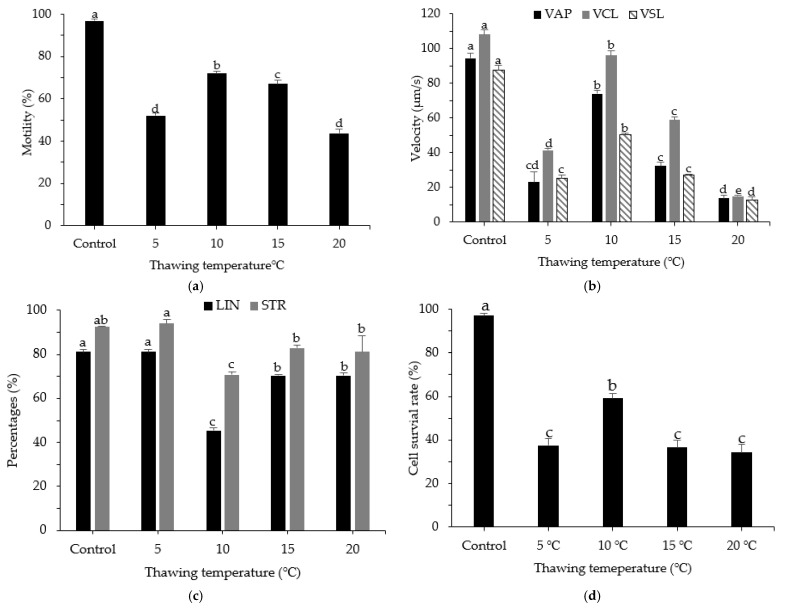
Computer-assisted sperm analysis of frozen and fresh sperm from roughscale sole, *Clidoderma asperrimum*, using various thawing temperatures. (**a**) Motility (%), (**b**) average path velocity (VAP; μm s^−1^), curvilinear velocity (VCL; μm s^−1^), straight-line velocity (VSL; μm s^−1^), (**c**) linearity (LIN; %), straightness (STR; %), and (**d**) cell survival rate (%). Different lowercase letters indicate significant differences among thawing temperatures (*p* < 0.05). Control, fresh sperm.

**Figure 6 animals-12-02553-f006:**
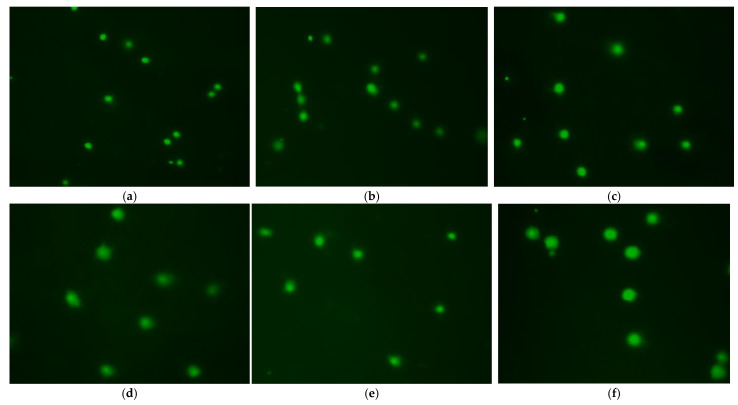
Fluorescent images of frozen and fresh sperm from roughscale sole, *Clidoderma asperrimum*, using various cryoprotective agents with 300 mM sucrose after comet assay. Each comet represents the damage level of DNA in sperm. (**a**) Fresh sperm, (**b**) Cryopreserved sperm using 10% DMSO, (**c**). Cryopreserved sperm using 10% propylene glycol, (**d**) Cryopreserved sperm using 10% ethylene glycol, (**e**) Cryopreserved sperm using 10% Methanol (**f**) Cryopreserved sperm using 10% Glycerol.

**Figure 7 animals-12-02553-f007:**
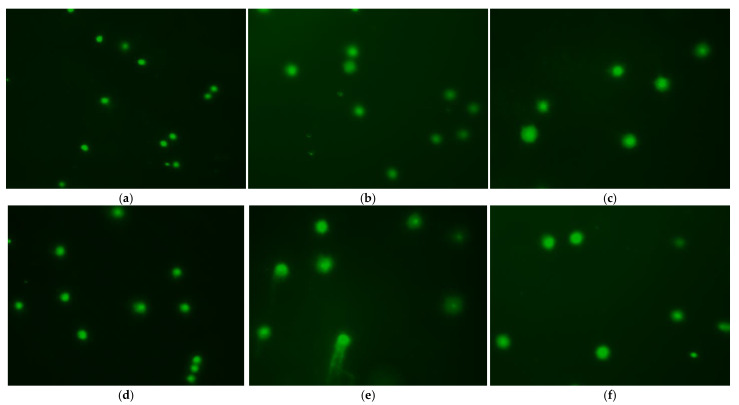
Fluorescent images of frozen and fresh sperm from roughscale sole, *Clidoderma asperrimum*, using various diluents with 10% DMSO as the cryoprotectant after comet assay. Each comet represents the damage level of DNA in sperm. (**a**) Fresh sperm, (**b**) Cryopreserved sperm using 300 mM Sucrose, (**c**) Cryopreserved sperm using 300 mM Glucose, (**d**) Cryopreserved sperm using Stain’s Solution, (**e**) Cryopreserved sperm using Ringer’s solution, (**f**) Cryopreserved sperm using Ringer’s solution.

**Figure 8 animals-12-02553-f008:**
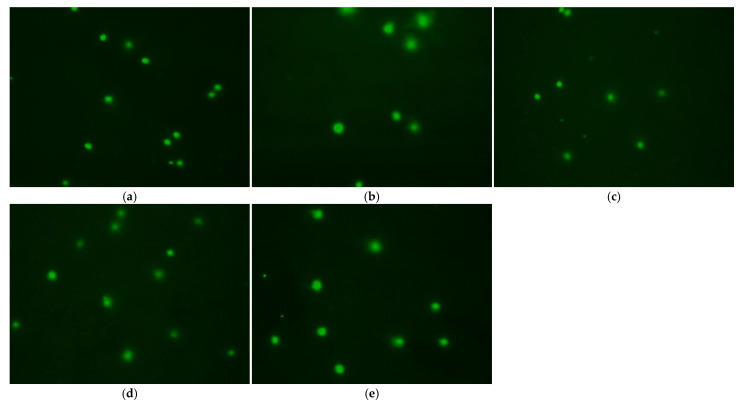
Fluorescent images of frozen and fresh sperm from roughscale sole, Clidoderma asperrimum, using various concentrations of DMSO with Stein’s solution as the diluent. (**a**) Fresh sperm, (**b**) Cryopreserved sperm using 5% DMSO, (**c**) Cryopreserved sperm using 10% DMSO, (**d**) Cryopreserved sperm using 15% DMSO, (**e**) Cryopreserved sperm using 20% DMSO.

**Figure 9 animals-12-02553-f009:**
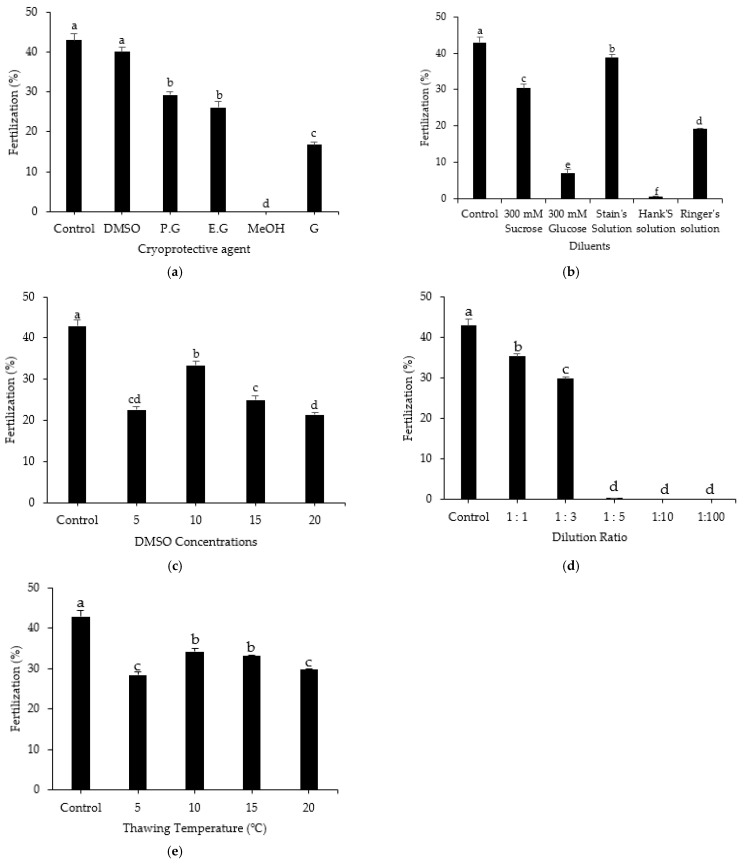
Fertilization rates of roughscale sole, *Clidoderma asperrimum,* eggs fertilized with fresh sperm and sperm cryopreserved with various treatment methods: (**a**) cryoprotective agent, (**b**) diluent, (**c**) DMSO concentration, (**d**) dilution ratio, and (**e**) thawing temperature. Different lowercase letters indicate significant differences among treatment methods (*p* < 0.05). Control, fresh sperm.

**Table 1 animals-12-02553-t001:** Statistical analysis of the comet assay performed after cryopreservation of roughscale sole (*Clidoderma asperrimum*) sperm using various cryoprotective agents (CPAs).

	Control	Different Cryoprotective Agent				
		DMSO	P.G	E.G	Methanol	Glycerol
Head Length	43.87 ± 0.385 ^c^	49.36 ± 0.501 ^a^	40.33 ± 0.487 ^d^	45.27 ± 0.414 ^b^	45.98 ± 0.489 ^b^	40.36 ± 0.380 ^d^
Tail Length	24.33 ± 0.219 ^e^	31.50 ± 0.296 ^d^	33.62 ± 0.346 ^c^	32.93 ± 0.253 ^c^	40.77 ± 0.280 ^a^	35.85 ± 0.270 ^b^
Head Intensity	94.25 ± 0.263 ^a^	88.18 ± 0.327 ^b^	83.23 ± 0.325 ^c^	82.81 ± 0.265 ^c^	74.48 ± 0.186 ^e^	78.71 ± 0.222 ^d^
Tail Intensity	5.70 ± 0.266 ^a^	11.75 ± 0.322 ^b^	16.50 ± 0.192 ^c^	17.19 ± 0.265 ^d^	25.52 ± 0.186 ^f^	21.29 ± 0.222 ^e^
% Tail DNA	1.305 ± 0.056 ^a^	3.319 ± 0.093 ^b^	5.948 ± 0.132 ^d^	4.295 ± 0.081 ^c^	4.488 ± 0.157 ^c^	6.691 ± 0.142 ^e^
Tail Migration	2.41 ± 0.117 ^a^	7.06 ± 0.186 ^b^	12.21 ± 0.145 ^c^	12.71 ± 0.084 ^d^	18.14 ± 0.109 ^f^	15.35 ± 0.137 ^e^

Different lowercase letters indicate significant differences among CPAs. Control: fresh sperm; PG: propylene glycol; EG: ethylene glycol.

**Table 2 animals-12-02553-t002:** Statistical analysis of the comet assay performed after cryopreservation of roughscale sole (*Clidoderma asperrimum*) sperm using various diluents.

	Control	Different Diluents				
		300 mM Sucrose	300 mM Glucose	Stain’s Solution	Hank’s Solution	Ringer’s Solution
Head Length	43.87 ± 0.385 ^c^	45.84 ± 0.432 ^a^	39.24 ± 0.350 ^c^	46.31 ± 0.444 ^a^	39.44 ± 0.406 ^c^	39.91 ± 0.391 ^c^
Tail Length	24.33 ± 0.219 ^e^	34.80 ± 0.239 ^d^	37.42 ± 0.221 ^b^	30.99 ± 0.278 ^e^	38.73 ± 0.238 ^a^	36.05 ± 0.233 ^c^
Head Intensity	94.25 ± 0.263 ^a^	86.18 ± 0.487 ^c^	70.03 ± 0.221 ^e^	89.20 ± 0.307 ^b^	68.82 ± 0.266 ^f^	72.70 ± 0.245 ^d^
Tail Intensity	5.70 ± 0.266 ^a^	13.82 ± 0.487 ^c^	29.97 ± 0.22 ^e^	10.80 ± 0.307 ^b^	31.18 ± 0.266 ^f^	27.30 ± 0.245 ^d^
% Tail DNA	1.305 ± 0.056 ^a^	3.696 ± 0.127 ^c^	8.402 ± 0.063 ^e^	2.835 ± 0.083 ^b^	8.941 ± 0.065 ^f^	7.529 ± 0.062 ^d^
Tail Migration	2.41 ± 0.117 ^a^	11.84 ± 0.12 ^c^	17.80 ± 0.102 ^e^	8.01 ± 0.126 ^b^	19.01 ± 0.099 ^f^	12.44 ± 0.072 ^d^

Different lowercase letters indicate significant differences among diluents. Control: fresh sperm.

**Table 3 animals-12-02553-t003:** Statistical analysis of the comet assay performed after cryopreservation of roughscale sole (*Clidoderma asperrimum*) sperm using various DMSO concentrations.

	Control	DMSO Concentrations (%)			
		5	10	15	20
Head Length	43.87 ± 0.385 ^c^	48.56 ± 0.349 ^b^	49.15 ± 0.370 ^b^	50.36 ± 0.397 ^a^	50.56 ± 0.436 ^a^
Tail Length	24.33 ± 0.219 ^e^	34.91 ± 0.208 ^b^	30.31 ± 0.211 ^d^	32.84 ± 0.226 ^c^	38.67 ± 0.220 ^a^
Head Intensity	94.25 ± 0.263 ^a^	86.92 ± 0.434 ^d^	92.11 ± 0.264 ^b^	90.46 ± 0.355 ^c^	84.50 ± 0.475 ^e^
Tail Intensity	5.70 ± 0.266 ^a^	13.08 ± 0.434 ^d^	7.89 ± 0.264 ^b^	9.54 ± 0.355 ^c^	15.50 ± 0.475 ^e^
% Tail DNA	1.305 ± 0.056 ^a^	3.612 ± 0.124 ^d^	2.095 ± 0.071 ^b^	2.626 ± 0.097 ^c^	4.711 ± 0.165 ^e^
Tail Migration	2.41 ± 0.117 ^a^	10.65 ± 0.098 ^b^	6.07 ± 0.067 ^b^	7.65 ± 0.100 ^c^	12.44 ± 0.072 ^f^

Different lowercase letters indicate significant differences among DMSO concentrations. Control: fresh sperm.

## Data Availability

The data presented in this study are available on request from the corresponding author.

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
