# Peer review of "Cryopreservation of Roughscale Sole (Clidoderma asperrimum) Sperm: Effects of Cryoprotectant, Diluent, Dilution Ratio, and Thawing Temperature"

_animals, 2022, doi:10.3390/ani12192553_

Round 1
Reviewer 1 Report
General comments:
In your manuscript entitled " Cryopreservation of roughscale sole (Clidoderma asperrimim) sperm: effects of cryoprotectant, diluent, dilution ratio, and thawing temperature" you present your study on optimizing cryopreservation conditions for roughscale sole sperm to be used in artificial reproduction in fish hatcheries.
Your aim is to establish an ideal cryopreservation protocol for roughscale sole sperm for commercial use in fisheries. Rationale being, that roughscale sole is caught only in relatively low numbers in regular fisheries, and that the meat of the roughscale sole is apparently light and elastic in character and thus of good quality for sashimi...
The study is well designed, well executed, the study parameters are well chosen and the results are clearly presented. The aim of the study, finding a better cryopreservation protocol for roughscale sperm, was achieved.
As a zoologist, I am a bit disappointed that the rationale of this study is only the fact, that you believe that not enough of this fish can be caught in the wild and that it is therefore necessary to enhance breeding for this species in commercial hatcheries in order to improve access to enough light and elastic fish meat for more sashimi...it is therefore a bit questionable if there really is a sound scientific interest and purpose here in your study or if this manuscript refers only to enhancing a breeding technique and thus lacks reals scientific merit. I am thus asking you to re-think, if this manuscript might not better be published in a journal focusing only on fisheries and fish rearing.
Specific comments:
None.
Author Response
Dear Reviewer
First of all, I would like to appreciate the constructive remarks that the reviewers gave for this manuscript. The authors have added information about conservation and genetic improvement of resources to strengthen the scientific value of this manuscript and make the research seem more logical. Sperm cryopreservation can make an important contribution to the germ storage of all transgenic lines. The benefit of this experiment is not only to support sperm banking for seed production in the aquaculture sector but also to preserve the roughscale sole (Clidoderma asperrimim) that was categorized as an endangered species. The International Union for Conservation of Nature's Red List of Threatened Species observed that the roughscale sole need for conservation (Tomiyama et al 2021). Through sperm cryopreservation research on this fish, it can help the program carried out by the Korean Fisheries government in the form of a stock enhancement program that releases hatchery-bred individuals into the wild.

Reviewer 2 Report
Roughscale sole is an endangered species and cryopreservation protocols are being investigated for the first time; therefore the research is scientifically sound, relevant, and original. The article is very well written; title and abstract reflect the contents of the paper, and introduction clearly states the problem being investigated and provides an adequate background. Methodology is well described, results are clearly presented, discussion is appropriated, and conclusions are supported by the results. My recommendation is to accept in the present form.
Author Response
Dear Reviewer
We would like to thank the Reviewer for the comment and for classification of the manuscript as very good and priority publishing and we are grateful for your consideration of this manuscript.

Reviewer 3 Report
This is an interesting and well written manuscript; the information is well presented, the language is used correctly and very easy to understand. However experimental design should be revised, because a small number of replicates were conducted.
Author Response
Dear Reviewer
Thank you very much for giving us the opportunity to improve and resubmit our manuscript. We would like to appreciate the constructive remarks that reviewers
gave for this manuscript.

Reviewer 4 Report
Dear authors,
please find my suggestion and recommendations which should be included in the presented manuscript "Cryopreservation of roughscale sole sperm: effects of cryoprotectant, diluent, dilution ration and thawing temperature" for being successfully accepted for publication.
Thank you for the excellent work in which I really appreciate final analysis of the fertilization rate and moreover practical impact of the developed protocol for Clidoderma asperrimum sperm cryopreservation.
Abstract
This part is generally well written but in
ln 31 - details about the supplier of the CASA system could be omitted. Moreover in the methodological part of the abstract information about other used methods missing. Please add it.
ln 29 - please change the word factors. Because you did not study factors but factors such as dilution, various types of CPs (i.e.factors) affect the parameters of spermatozoa.
And finally, in the introduction part of the abstract please indicate why have you chosen the roughscale sole spermatozoa for your study.
Introduction
change the word seed in whole text of the manuscript
ln 69 - check the use of the word harm
ln 79 - I really recommend to authors to change the word purpose to e.g. The main objective or The main aim of our study etc.
Material and methods
ln 86 - change to word observation
ln 93 - please add the citation to support dosage of hormonal treatment by Ovaprim.
ln 105 - why did the authors choose 300 mM sucrose solution for testing different kinds of CPAs
ln 105 - why did the authors choose 10% concentration of all CPA`s?
why did the authors choose also methanol for testing? Does it commonly use for fish sperm cryopreservation?
ln 107 - does the cryopreservation of fish sperm is usually without the equilibration period? Support with reference.
ln 116 - please add citation to support the method of thawing.
lns 117-118 - detailed composition of the media must be added
subsection 2.3. - did authors analyze sperm concentration of samples before testing sperm : extender dilution ratio effect?
ln 130 - Why did the authors use different temperatures for thawing in this experiment?
ln 137 - please correct the product name of chambers used for CASA analysis
ln 138 - I think that magnification with eyepieces is 100x. correct it
ln 139 - specify FPS od camera
ln 141 - specify treshold values for total motility
ln 147 - indicate temperature used during recording the sperm motility
ln 170 - put better description of used formula. What does DNAc stand for? Describe better the measured parameters. They must be in consequence with presented results in tables 1-3-
ln 172 - check the correct use of the word injury
ln 174 - add the citation to support method for the fertilization ability analyzes
Results
The results are well presented however I recommend moving subsection 3.7. before subsection 3.6. It will have more logical consequences.
If authors have images from the Comet assay, they can be added as a representative example. This will increase the attractiveness of paper.
In the legend of the Table 1 - 3 I found the term fresh milt I recommend authors to keep the same name for control group - fresh sperm as indicated in the previous figures.
Discussion
ln 344 - change the adjective good to a more suitable word for a scientific manuscript.
ln 368 - The fact that CPAs are species-dependent is true but I will be more careful about this statement. According to your results you can not write that directly since you probably used e.g. different kind of extender composition.
lns 423-425 in this sentence do authors mean by word concentration - volume? if not rewrite the sentence.
Is there any effect of removing seminal plasma before cryopreservation in fish species?
ln 427 - change to word melting to thawing
ln 441 - In the case of studies related to the protocols for cryopreservation of sperm, their DNA integrity analysis is mainly used for certain level of fertilizing ability prediction.
ln 475 - change the at least to the minimum DNA damage
My best regards
Author Response
Dear Reviewer
We would like to further express our sincere thanks for the helpful comments from the reviewers. We have revised the manuscript according to reviewer comments. The comments have been very thorough and useful in improving the manuscript. We strongly believe that the comments and suggestions have increased the scientific value of revised manuscript.
